# Chr15q25 Genetic Variant rs16969968 Alters Cell Differentiation in Respiratory Epithelia

**DOI:** 10.3390/ijms22136657

**Published:** 2021-06-22

**Authors:** Zania Diabasana, Jeanne-Marie Perotin, Randa Belgacemi, Julien Ancel, Pauline Mulette, Claire Launois, Gonzague Delepine, Xavier Dubernard, Jean-Claude Mérol, Christophe Ruaux, Philippe Gosset, Uwe Maskos, Myriam Polette, Gaëtan Deslée, Valérian Dormoy

**Affiliations:** 1Inserm UMR-S1250, P3Cell, University of Reims Champagne-Ardenne, SFR CAP-SANTE, 51092 Reims, France; zania.diabasana@inserm.fr (Z.D.); jmperotin-collard@chu-reims.fr (J.-M.P.); randa.belgacemi@lundquist.org (R.B.); jancel@chu-reims.fr (J.A.); pmulette@chu-reims.fr (P.M.); gdelepine@chu-reims.fr (G.D.); jcmerol@chu-reims.fr (J.-C.M.); myriam.polette@univ-reims.fr (M.P.); gdeslee@chu-reims.fr (G.D.); 2Department of Respiratory Diseases, CHU de Reims, Hôpital Maison Blanche, 51092 Reims, France; claunois@chu-reims.fr; 3Department of Thoracic Surgery, CHU de Reims, Hôpital Maison Blanche, 51092 Reims, France; 4Department of Otorhinolaryngology, CHU de Reims, Hôpital Robert Debré, 51092 Reims, France; xdubernard@chu-reims.fr; 5Department of Otorhinolaryngology, Clinique Mutualiste La Sagesse, 35043 Rennes, France; christophe.ruaux@hospigrandouest.fr; 6CNRS UMR9017, Inserm U1019, University of Lille, Centre Hospitalier Régional Universitaire de Lille, Institut Pasteur, CIIL—Center for Infection and Immunity of Lille, 59000 Lille, France; philippe.gosset@pasteur-lille.fr; 7Integrative Neurobiology of Cholinergic Systems, Institut Pasteur, CNRS UMR 3571, 75015 Paris, France; uwe.maskos@pasteur.fr; 8Department of Biopathology, CHU de Reims, Hôpital Maison Blanche, 51092 Reims, France

**Keywords:** rs16969968, airway epithelial cells, differentiation, remodelling, inflammation

## Abstract

The gene cluster region, CHRNA3/CHRNA5/CHRNB4, encoding for nicotinic acetylcholine receptor (nAChR) subunits, contains several genetic variants linked to nicotine addiction and brain disorders. The CHRNA5 single-nucleotide polymorphism (SNP) rs16969968 is strongly associated with nicotine dependence and lung diseases. Using immunostaining studies on tissue sections and air-liquid interface airway epithelial cell cultures, in situ hybridisation, transcriptomic and cytokines detection, we analysed rs16969968 contribution to respiratory airway epithelial remodelling and modulation of inflammation. We provide cellular and molecular analyses which support the genetic association of this polymorphism with impaired ciliogenesis and the altered production of inflammatory mediators. This suggests its role in lung disease development.

## 1. Introduction

Chronic inflammatory pulmonary diseases are characterized by alterations in lung tissues (e.g., airway remodelling, inflammation, fibrosis) and pulmonary function. Those alterations can be caused by the inhalation of noxious particles, including tobacco smoke. In addition to environmental factors, multiple genetic susceptibilities have been associated with the development of pulmonary diseases [1]. Nonetheless, only alpha1-antitrypsin deficiency provided insights into the pathogenesis of respiratory disorders and helped improve patient management [2]. The CHRNA3/CHRNA5/CHRNB4 gene cluster, encoding for the α3, α5 and β4 nicotinic acetylcholine receptor (nAChR) subunits, has drawn interest due to its implication in nicotine dependence and lung cancer. The activation of nAChRs by nicotine has been implicated in the modulation of airway epithelial cell (AEC) response and inflammatory reaction [3].

The non-synonymous single-nucleotide CHRNA5 polymorphism, rs16969968, has been identified as clinically significant [4,5,6]. This mutation is located in the chromosomal region 15q25.1 and, due to a G to A base change, leading to the amino acid substitution D398N (aspartic acid to asparagine) in the resulting protein [7,8]. Rs16969968, originally identified as a candidate gene associated with smoking [9], is now identified as a candidate gene associated with nicotine dependence [10], schizophrenia [11], and lung cancer [12,13,14], as well as during chronic obstructive pulmonary disease (COPD) and altered lung functions [15,16,17,18,19,20,21]. It was present in approximately 60% of the population and its frequency significantly increased in lung diseases [6,9,12,13,14,16,17]. Although the expression of the rs16969968 polymorphism (α5SNP) affected nicotine-induced signalling, its effects on lung histomorphological patterns are not established. In this study, we aimed to determine whether α5SNP is involved in AEC differentiation and remodelling in human lung.

## 2. Results

We first confirmed the detection of CHRNA5 expression in bronchial epithelial cells using in situ hybridisation on formalin-fixed paraffin-embedded (FFPE) lung tissues (Appendix A) [22]. We next compared epithelial remodelling features on lung tissues obtained from α5WT and α5SNP patients. Epithelial height did not differ between the two groups (Figure 1A,B). The percentage of remodelled epithelium, including goblet cell hyperplasia, basal cell hyperplasia, and metaplasia, as well as the proliferative index, did not show any significant difference between the two groups (Figure 1A–D). Focusing on the main bronchial epithelial cell populations, we observed that the number of multiciliated cells was significantly decreased in α5SNP patients compared to α5WT patients (247.4 ± 15.93 ciliated cells/mm vs. 186.9 ± 8.67 ciliated cells/mm, *p* < 0.01) (Figure 1C,D). There was no difference in the number of basal cells or secretory cells (Muc5ac-, Muc5b-, and Uteroglobin-secreting cells) between the two groups.

To dissect the role of rs16969968 during differentiation, we performed a kinetic analysis of AEC cultured in ALI conditions for 35 days. Transepithelial electrical resistance was significantly higher in α5SNP compared to the α5WT group, as observed at ALI35 (325.1 ± 16.98Ω/cm^2^ vs. 169.3 ± 17.55Ω/cm^2^, *p* < 0.01) (Figure 2A). Focusing on multiciliated, basal, and mucin-secreting cells, we performed comparative quantitative transcriptomic and localization analyses during AEC differentiation. Transcript analyses at ALI-7, ALI-14, and ALI-35 demonstrated a significant decrease of 40 to 85% in FOXJ1, CK5, MUC5AC, and MUC5B gene expressions in α5SNP compared to α5WT cells (Figure 2B). α5SNP-ALI cultures had fewer multiciliated cells at ALI-14 (3694 ± 294.0 mean grey value (MGV) in α5WT vs. 2681 ± 302.6 MGV in α5SNP-expressing cells, *p* < 0.05) and this pattern was maintained at the end of differentiation, although the difference was not significant (11160 ± 1081 MGV in α5WT vs. 9641 ± 620.8 MGV in α5SNP-expressing cells, *p* = 0.21) (Figure 2C,D). α5SNP-ALI cultures had more basal cells at ALI35 (2950 ± 752.6 MGV in α5WT vs. 4672 ± 526.6 MGV in α5SNP-expressing cells) but did not reach statistical significance (Figure 2C,D). The number of mucin-secreting cells did not significantly differ between the two groups (Figure 2C,D).

We next analysed the expression of inflammatory mediators in apical and basal cell culture supernatants (Figure 3 and Appendix A). On average, there was a significant two-fold decrease in pro- and anti-inflammatory protein expression in α5SNP-expressing cells compared to α5WT, including GR0α, sICAM-1, IFNγ, IL-2/5/6/10/12p70/32α, IP-10, MCP-1, MIF, RANTES, and SDF-1. In addition, IL-17/17E/1ra/1α/1β/4, and MIP-1α were also detected but the ratios did not statistically differ. The ratios were consistent for both secretory compartments.

## 3. Discussion

Our data provide experimental clues on AEC to fill the knowledge gap in our understanding of the rs16969968 polymorphism and its genetic association with lung diseases [16]. We demonstrated that α5SNP expression was associated with structural and functional alterations of airway epithelium. One of the most important findings is the α5SNP-associated alteration of ciliogenesis, which we observed in both bronchial epithelia and AEC ALI cultures. Cell differentiation was deregulated, as evidenced by transcript levels of ciliated, basal, and mucin-secreting cells. In addition, basal cell cytokine production was reduced, suggesting a global impairment of the epithelial contribution to the inflammatory response during epithelial cell differentiation.

We focused here on the respiratory epithelium from patients with no chronic inflammatory pulmonary disease to investigate the impact of rs16969968 as a forerunner of pulmonary dysregulation. Expanding the experimental approaches to small airways and lung diseases will allow us to fully decipher the role of rs16969968 polymorphism susceptibility in lung histological features [23]. Furthermore, besides epithelial cells, the recruitment of inflammatory cell populations and their cytokine profiles should also be evaluated in α5SNP patients at baseline and in response to an inflammatory stimulus.

Although rs16969968 was identified and associated with several major diseases more than a decade ago, very few molecular studies tackled the biological relevance of this genetic alteration. It has been only demonstrated that nAChRs containing rs16969968-α5 subunit exhibited an altered response to agonists and calcium permeability [24,25,26]. Considering the increase in the number of studies reporting its significant genetic association, the molecular and cellular dissection of rs16969968-associated pathogenic mechanisms is paramount.

In conclusion, using a combination of in vitro and ex vivo approaches, we have obtained compelling evidence in support of rs16969968, directly inducing molecular and cellular changes in AEC, ultimately leading to the remodelling and the impairment of the epithelial-related immune response of the airways.

## 4. Materials and Methods

### 4.1. Human Subjects

Patients scheduled for lung resection for cancer (University Hospital of Reims, Reims, France) were prospectively recruited (*n* = 16) following the standards established and approved by the institutional review board of the University Hospital of Reims, France (IRB Reims-CHU, approval date: 2011/06/12). Informed consent was obtained from all the patients. Patients with chronic obstructive pulmonary disease, asthma, cystic fibrosis, bronchiectasis, or pulmonary fibrosis were excluded. At inclusion, age, sex, smoking history, and pulmonary function test results were recorded.

### 4.2. Human Primary Airway Epithelial Cell Cultures

Human primary airway epithelial cells (AEC) were obtained from nasal polyps resected from non-COPD patients (13 donors) to establish air-liquid interface (ALI) cultures, as described by us and others [27,28,29,30,31,32]. Cells were dissociated by overnight pronase incubation (0.5 mg/mL, Sigma-Aldrich, Saint Quentin Falavier, France) and counted with ADAM (NanoEnTek, VWR, Fontenay-sous-Bois, France) according to NanoEnTek instructions. In total, 200,000 cells were seeded on 12-well plates containing 0.4 µm Transwells (Corning, Fisher Scientific, Illkirch, France) coated with 0.3 mg/mL collagen type IV from the human placenta (Sigma-Aldrich, Saint Quentin Falavier, France). PneumaCult-EX (PnC-Ex) media (StemCell, Saint-Egrève, France) were used for initial proliferation in apical and basal chambers. Upon reaching cell confluency, the apical medium was removed, and PneumaCult-ALI (PnC-ALI, StemCell, Saint-Egrève, France) medium was used in the basal chamber. The culture medium was changed three times a week and cells were kept in incubators at 37 °C, 5% CO2. Cells and supernatants were collected every 7 days to generate kinetic analysis. Cell morphological changes were evaluated weekly under a microscope.

### 4.3. TEER Measurements

Transepithelial electrical resistance (TEER) was evaluated every 7 days using an EVOM2 resistance meter with an STX2 electrode (World Precision Instruments Ltd, Hitchin, UK) at room temperature. The electrode was equilibrated in PnC-ALI for 1 h at room temperature before measurement. One mL PnC-ALI was added to the apical compartment and triplicate measurements were performed per sample. Data were corrected for blank values and area. Average resistance was subtracted from the measured value of every well according to data acquired on cell-free permeable supports and results are presented as resistance per surface (Ω.cm^2^).

### 4.4. RT-qPCR Analyses

Total RNA from AEC was isolated by High Pure RNA isolation kit (Roche Diagnostics, Meylan, France) and 250 ng was reverse transcribed into cDNA by Transcriptor First Stand cDNA Synthesis kit (Roche Diagnostics, Meylan, France). Quantitative PCR reactions were performed with the fast start universal probe master kit and UPL-probe system in a LightCycler 480 Instrument (Roche Diagnostics, Meylan, France) as recommended by the manufacturer. Primers used are listed in Appendix A. Results for all expression data regarding transcripts were normalized to the expression of the house-keeping gene, GAPDH, amplified with the following primers: forward 5′-ACCAGGTGGTCTCCTCTGAC-3′, reverse 5′-TGCTGTAGCCAAATTCGTTG-3′. Relative gene expression was assessed by the ΔΔCt method and expressed as 2-ΔΔCt. The transcript levels of α5WT mRNAs were normalized at 1.00 and the transcript levels of SNP mRNAs were comparatively assessed for each gene and at each time-point.

### 4.5. DNA Extraction

Epithelial cell pellets obtained from dissociated human polyps (GenElute™ RNA/DNA/Protein Plus Purification Kit, Sigma-Aldrich, Saint Quentin Falavier, France) or four tissue sections (20 µm of thickness each) trimmed from formalin-fixed paraffin-embedded (FFPE) lung tissue blocks (GenElute™ FFPE RNA/DNA Purification Plus Kit) were processed for DNA purification according to the manufacturer’s instructions.

### 4.6. Genotyping

nAChR α5 subunit coding gene, CHRNA5, was amplified with DNA polymerase TaKaRa LA Taq (TAKARA Bio Europe SAS, Saint-Germain-en-Laye, France) using the following primers: forward 5′- AGTCATGTAGACAGGTACTTCACTCAG-3′, reverse 5′- TGGAAGAAGATCTGCATTTG-3′. Amplification products were digested with Taq1 enzyme recognizing the following sequence: 5′-TCGA-3′, only present in the α5WT sequence. Digestion products were then separated by agarose gel electrophoresis and gels were imaged using a LAS-1000 Imager for analysis (Aïda software, Raytest, Courbevoie, France).

### 4.7. Immunohistochemistry (IHC) and Immunofluorescent (IF) Staining

Immunohistochemistry and immunofluorescent staining were performed on FFPE lung tissues distant from the tumour, as previously described [33]. Then, 5 μm sections were processed for hematoxylin and eosin staining and analysed on a microscope (x20) to assess epithelium height and bronchial epithelium remodelling. FFPE lung tissue section slides were deparaffinized and blocked with 10% BSA in PBS for 30 min at room temperature. Tissue sections were then incubated with the primary antibodies, as listed in Appendix A for one night at 4 °C in 3% BSA in PBS. After PBS wash, a second primary antibody was used for 2 h at room temperature to highlight non-differentiated cells, secretory cells, and intermediate filaments on epithelia: p63 (AF1916, R&D Systems, Bio-Techne SAS, Noyal Châtillon sur Seiche, France), Muc5ac (NBP2-15196, Novus Biological, Bio-Techne SAS, Noyal Châtillon sur Seiche, France), Muc5b (E-AB-15988, ElabScience, CliniSciences SAS, Nanterre, France), and Vimentin (M0725, Agilent Dako, Les Ulis, France). Sections were washed with PBS and incubated with the appropriate secondary antibodies in 3% BSA in PBS for 30 min at room temperature. DNA was stained with DAPI during incubation with the secondary antibodies. Micrographs were acquired on a Zeiss AxioImageur (x20 Ph) with ZEN software (8.1, 2012) and processed with ImageJ (National Institutes of Health) for analysis. For each patient, five random fields per section containing bronchi were taken to quantify cell proliferation, differentiated and non-differentiated cell expression, and mucus secretion on the bronchial epithelium. For each field, a threshold was established by subtracting the background with a rolling ball radius of 50.0 pixels setting the minimum at 0.

### 4.8. Whole-Mount Immunofluorescent Immunostaining (WMIF)

Methanol-fixed AEC from ALI cultures were rehydrated by decreasing methanol concentration before post-fixation with acetone. Cells were then blocked with 10% BSA in PBS for 2 h at room temperature and incubated for one night at 4 °C in 3% PBS/BSA in PBS with the primary antibodies as listed in Appendix A. Cells were washed with PBS and incubated with the appropriate secondary antibodies in PBS for 2 h at room temperature. DNA was stained with DAPI during incubation with the secondary antibodies. Clarification of cells was achieved by a glycerol gradient (25%/50%/75%) before mounting the slides. Micrographs were acquired on a Zeiss AxioImageur (x 20 Ph) with ZEN software (version 8.1, Zeiss, Marly le Roi, France) and processed with ImageJ, as previously described [34].

### 4.9. In Situ Hybridisation

In situ hybridisation was performed on FFPE lung tissue sections to assess CHRNA5 mRNA expression. Tissue sections were deparaffinized and pretreated with hydrogen peroxide. Target retrieval was then carried out in a steamer for 15 min and slides were dried overnight at room temperature. After protease pre-treatment, tissue sections were incubated with a CHRNA5 probe for 2 h at 40 °C. Target amplification was then achieved by successive hybridizations and the signal was detected using DAB substrates. The slides were counterstained using hematoxylin and dehydrated by alcohol and xylene baths before mounting. Micrographs were acquired on a Nikon Eclipse with NIS-Elements software and processed with ImageJ.

### 4.10. Immunoblot Analyses

Thirty-six human cytokines and chemokines expression in apical and basal chamber supernatants (500 μL) were assayed by proteome profiler array according to the R&D systems instructions (ARY005B). Final detection was obtained by streptavidin-HRP and chemiluminescence. Membranes were then imaged using ImageQuantTM LAS-4000 (GE Healthcare) for analyses. Detected signals were digitally quantified using ImageJ. The values were normalized to the positive and negative controls for each membrane and the proteins were considered detected if their mean maximum grey pixel value of detection exceeded the mean maximum grey pixel value of the negative controls by 5%.

### 4.11. Statistics

The data are expressed as mean values and percentages. A non-parametric Mann-Whitney test was used to analyse differences between groups or the one-sample t-test to the hypothetical value of 1.00 representing α5WT subjects. A *p*-value < 0.05 was considered significant.

## Figures and Tables

**Figure 1 ijms-22-06657-f001:**
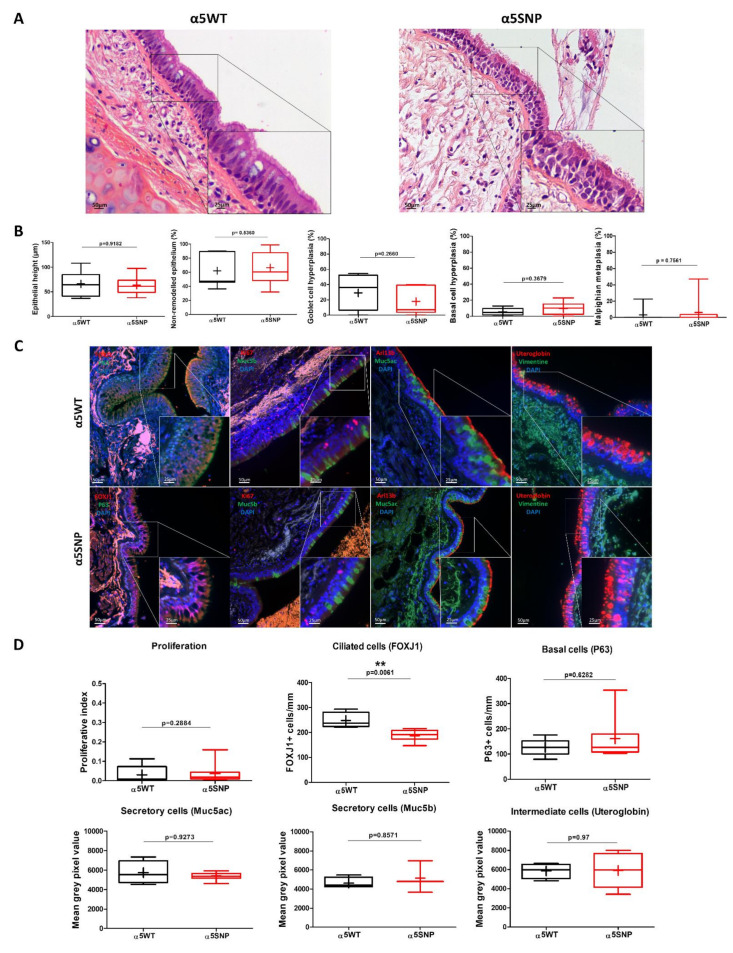
SNP rs16969968 is associated with a decrease in multiciliated cells in bronchi epithelia. (**A**) Representative micrographs showing the bronchial epithelia of α5WT- and α5SNP-expressing patients stained for haematoxylin and eosin. Magnification corresponding to the selected area is shown. (**B**) Box and whisker plots (median with min to max, the plus sign (+) represents the mean value) represent the measurements of epithelium height (µm), and the percentage of non-remodelled and remodelled (basal cell hyperplasia, goblet cell hyperplasia and metaplasia) epithelium in α5WT (black) and α5SNP-expressing (red) groups (n ≥ 7 for each group). (**C**) Representative micrographs showing the bronchial epithelia of α5WT and α5SNP-expressing patients stained for ciliated cells (Arl13b and FOXJ1, red), intermediate cells (uteroglobin, red), proliferative cells (Ki67, red), basal cells (P63, green), mucin-secreting cells (Muc5ac and Muc5b, green), intermediate filaments (vimentin, green) and cell nuclei (DAPI, blue). Magnification corresponding to the selected area is shown. (**D**) Box and whisker plots (median with min to max, the plus sign (+) represents the mean value) report the proportion of FOXJ1- and P63-positive cells per mm, the proliferative index, and the mean grey pixel values of mucin- and uteroglobin-associated fluorescence in α5WT (black) and α5SNP-expressing cells (red) (n ≥3 for each group). ** *p* < 0.01 α5WT vs. α5SNP.

**Figure 2 ijms-22-06657-f002:**
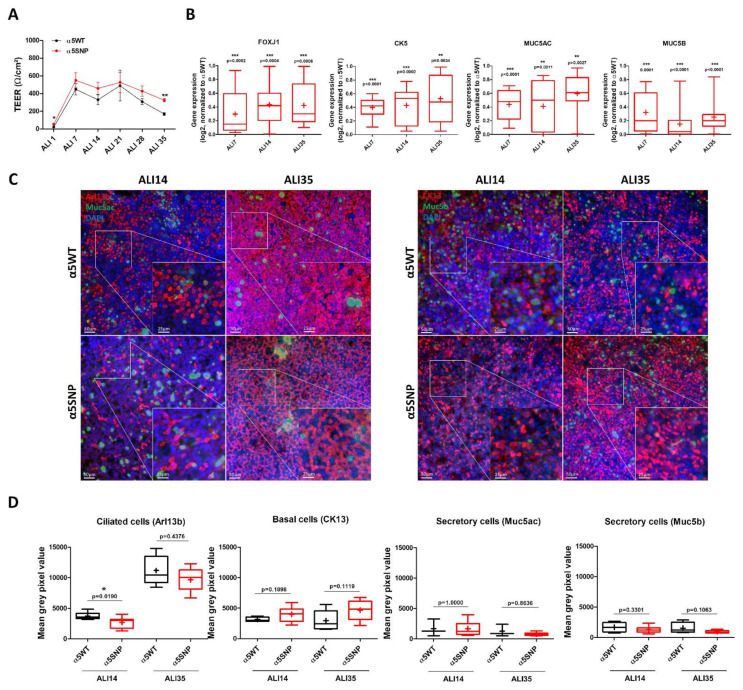
SNP rs16969968 alters respiratory airway epithelial cell differentiation. (**A**) Curves represent the TEER of AEC (n = 13) measured at ALI-1, ALI-7, ALI-14, ALI-21, ALI-28, and ALI-35 in α5WT (black) and α5SNP-expressing (red) cells. Means ± SEM are shown for each ALI time point. * *p* < 0.05; *** *p* < 0.001 α5WT vs. α5SNP. (**B**) Box and whisker plots (median with min to max, the plus sign (+) represents the mean value) represent the relative mRNA levels during ALI cultures by RT-qPCR (*n* = 13) for differentiation markers (CK5, non-differentiated cells; FOXJ1, ciliated cells; MUC5AC/MUC5B, mucous-secreting cells) in α5SNP-expressing groups. * *p* < 0.05; ** *p* < 0.01; *** *p* < 0.001 α5WT vs. α5SNP. (**C**) Examples of micrographs taken from AEC cultures at ALI-14 and ALI-35 showing multiciliated (Arl13b, red) and mucin-secreting (Muc5ac, green) cells on the left, basal (CK13, red) and mucin-secreting (Muc5b, green) cells on the right. Nuclei are stained in blue (DAPI). Magnification corresponding to the selected area is shown. (**D**) Box and whisker plots (median with min to max, the plus sign (+) represents the mean value) represent the mean grey pixel values of Arl13b-, CK13-, muc5ac- and muc5b-associated fluorescence at ALI-14 and ALI-35 in α5WT (black) and α5SNP-expressing cells (red) (n ≥ 3 for each group). * *p* < 0.05 α5WT vs. α5SNP.

**Figure 3 ijms-22-06657-f003:**
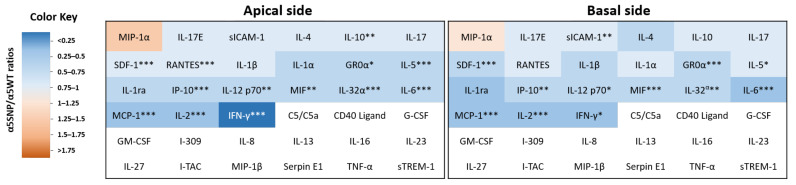
Cytokine and chemokine expression in AEC culture supernatants. Heat map colours correspond to the α5SNP/α5WT cytokine and chemokine ratios at ALI-7 from apical (**left**) and basal (**right**) supernatants obtained from AEC cultures (*n* = 4 α5WT and *n* = 9 α5SNP). Downregulated (blue) and upregulated (orange) proteins in supernatants from α5SNP-expressing cells are represented. Non-detected proteins are represented in white. * *p* < 0.05; ** *p* < 0.01; *** *p* < 0.001 α5WT vs. α5SNP.

## Data Availability

The data presented in this study are available on request from the corresponding author.

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
