# Peer review of "Chr15q25 Genetic Variant rs16969968 Alters Cell Differentiation in Respiratory Epithelia"

_ijms, 2021, doi:10.3390/ijms22136657_

Round 1

Reviewer 1 Report

A very well done structured study. It is of both methodological and scientific interest for the readers of the journal. Can serve as a model for planning an appropriate study design. From the point of view of scientific significance, it is valuable in obtaining molecular biological data on the genetic association of single nucleotide polymorphism (SNP) rs16969968 with impaired ciliogenesis and altered production of inflammatory mediators. The results obtained indicate the role of this polymorphism in the development of lung diseases, which has a scientific novelty.

Author Response

We thank the reviewer for his evaluation of our study and his kind remarks.

Reviewer 2 Report

The communication entitled „Chr15q25 genetic variant rs16969968 alters cell differentiation

in respiratory epithelia” presents cellular and molecular analyses of polymorphism rs16969968 and its role in lung diseases development.

In general the manuscript is factual, straightforward, clear and informative. The manuscript may be interested for readers of the Journal.

My questions/suggestions

  1. Line 21 – detail the gene for rs16969968 SNP in Abstract section.
  2. Line 48 – COPD, expand the shortcut, please
  3. I am wonder, if there are any data about rs16969968 polymorphism frequencies (alleles/genes) in lung disease patients?

Author Response

We thank the reviewer for his evaluation of our study and his kind remarks.

Concerning the suggestions, we modified the manuscript accordingly:

  1. We added CHRNA5 for the SNP of interest in the abstract
  2. We expanded COPD since it was only mentioned in the abbreviations
  3. Rs16969968 frequencies in chronic lung diseases were reported in the original GWAS mentioned in the manuscript. Interestingly, a recent study has investigated the association between rs16969968 polymorphism and lung function, smoking status and pulmonary diseases in Non-Hispanic white subjects (Ref#16 in the manuscript). In this cohort from the National Lung Screening Trial (NLST) including 9270 patients:
    1. COPD subjects (n=674) included 16% of homozygous minor allele (AA), 50% of heterozygous (AG) and 34% of homozygous major allele (GG);
    2. Asthmatic subjects (n=636) included 13% of AA, 47% of AG and 40% of GG.
    3. Lung cancer patients (n=373) included 16% of AA, 47% of AG and 37% of GG.

Overall, this study re-confirmed CHRNA5 rs16969968 polymorphism association with COPD and lung cancer while detailing the repartition of the alleles. We added a comment at the end of the introduction.